# Green Synthesis of Silver Nanoparticles from *Allium cepa* L. Peel Extract, Their Antioxidant, Antipathogenic, and Anticholinesterase Activity

**DOI:** 10.3390/molecules28052310

**Published:** 2023-03-02

**Authors:** Mehmet Fırat Baran, Cumali Keskin, Ayşe Baran, Abdulkerim Hatipoğlu, Mahmut Yildiztekin, Selçuk Küçükaydin, Kadri Kurt, Hülya Hoşgören, Md. Moklesur Rahman Sarker, Albert Sufianov, Ozal Beylerli, Rovshan Khalilov, Aziz Eftekhari

**Affiliations:** 1Department of Food Processing, Vocational School of Technical Sciences, Batman University, Batman 72100, Turkey; 2Department of Biology, Mardin Artuklu University Graduate Education Institute, Mardin 47200, Turkey; 3Department of Nutrition and Dietetics, Faculty of Health Sciences, Mardin Artuklu University, Mardin 47200, Turkey; 4Department of Herbal and Animal Production, Köyceğiz Vocational School, Mugla Sitki Kocman University, Mugla 48000, Turkey; 5Department of Medical Services and Techniques, Köyceğiz Vocational School of Health Services, Mugla Sitki Kocman University, Mugla 48000, Turkey; 6Alternative Energy Resources Technology Program, Department of Electricity and Energy, Beşiri Organized Industrial Zone Vocational School, Batman 72100, Turkey; 7Department of Biology, Dicle University Faculty of Sciences, Diyarbakır 21280, Turkey; 8Department of Pharmacy, State University of Bangladesh, 77 Satmasjid Road, Dhanmondi, Dhaka 1205, Bangladesh; 9Educational and Scientific Institute of Neurosurgery, Peoples’ Friendship University of Russia (RUDN University), 6 Miklukho-Maklaya St., 117198 Moscow, Russia; 10Department of Neurosurgery, Sechenov First Moscow State Medical University (Sechenov University), 119992 Moscow, Russia; 11Central Research Laboratory, Bashkir State Medical University, 450008 Ufa, Russia; 12Department of Biophysics and Biochemistry, Baku State University, AZ1148 Baku, Azerbaijan; 13Department of Biochemistry, Faculty of Science, Ege University, Izmir 35040, Turkey

**Keywords:** *Allium cepa*, anticholinesterase, antioxidant, antibacterial, green synthesis, silver nanoparticles

## Abstract

The present work deals with the green synthesis and characterization of silver nanoparticles (AgNPs) using *Allium cepa* (yellowish peel) and the evaluation of its antimicrobial, antioxidant, and anticholinesterase activities. For the synthesis of AgNPs, peel aqueous extract (200 mL) was treated with a 40 mM AgNO3 solution (200 mL) at room temperature, and a color change was observed. In UV-Visible spectroscopy, an absorption peak formation at ~439 nm was the sign that AgNPs were present in the reaction solution. UV-vis, FE-SEM, TEM, EDX, AFM, XRD, TG/DT analyses, and Zetasizer techniques were used to characterize the biosynthesized nanoparticles. The crystal average size and zeta potential of AC-AgNPs with predominantly spherical shapes were measured as 19.47 ± 1.12 nm and −13.1 mV, respectively. Pathogenic microorganisms *Bacillus subtilis*, *Staphylococcus aureus*, *Escherichia coli*, *Pseudomonas aeruginosa*, and *Candida albicans* were used for the Minimum Inhibition Concentration (MIC) test. When compared to tested standard antibiotics, AC-AgNPs demonstrated good growth inhibitory activities on *P. aeuruginosa*, *B. subtilis*, and *S. aureus* strains. In vitro, the antioxidant properties of AC-AgNPs were measured using different spectrophotometric techniques. In the β-Carotene linoleic acid lipid peroxidation assay, AC-AgNPs showed the strongest antioxidant activity with an IC_50_ value of 116.9 µg/mL, followed by metal-chelating capacity and ABTS cation radical scavenging activity with IC_50_ values of 120.4 µg/mL and 128.5 µg/mL, respectively. The inhibitory effects of produced AgNPs on the acetylcholinesterase (AChE) and butyrylcholinesterase (BChE) enzymes were determined using spectrophotometric techniques. This study provides an eco-friendly, inexpensive, and easy method for the synthesis of AgNPs that can be used for biomedical activities and also has other possible industrial applications.

## 1. Introduction

Nanobiotechnology advancements have resulted in intriguing discoveries in materials science. Biosynthesis of economically and environmentally beneficial metal nanoparticles can now be done using this technology in the fields of cosmetics, environmental safety, defense, food, agriculture, and health [1]. Various plant components including leaves, roots, and fruits are generally used for biosynthesis applications, also called “green synthesis” [2,3,4,5,6]. Traditional chemical and physical procedures can also be used to produce nanomaterials. However, chemical approaches have drawbacks such as biotoxicity for the environment and humans, while physical ones require a great deal of energy. Furthermore, physically manufactured nanoparticles lack the appropriate morphology, size, composition, crystallinity, size distribution, and shape distribution [7].

There are several metals and metal-based oxide nanosystems that have been created and used in the catalytic and antibacterial industries. Silver nanoparticles (AgNPs) have received particular attention among the many types of nanoparticles because of their unique physiochemical and optoelectronic characteristics, which make them effective fungicidal, bactericidal, anticancer, and catalytic agents. AgNPs’ antibiotic resistance capabilities against multidrug-resistant bacteria also make them more esteemed. *Staphylococcus aureus* and *Escherichia coli* are only two of the pathogenic pathogens that AgNPs show potent antibacterial activities against [4,8,9].

Remarkably, silver nanoparticles (Ag-NPs) have a narrow plasmon resonance, a high surface-to-volume ratio, unique physicochemical features, and a variety of uses in microelectronics, biology, and medical research. Due to their widespread application in numerous economic and pharmacologically relevant products, AgNPs have attracted a great deal of interest compared to other metal NPs. The conventional techniques for synthesis, such as physical, thermal, hydrothermal, and chemical modes, are pricy and exceedingly risky and rely on harmful substances. Hence, a green synthesis strategy utilizing biological resources for the effective formation of NPs is the focus. The basis of this environmentally friendly process is the synthesis of green nanoparticles using chemicals that are both ecologically safe and renewable as reducing and capping agents. Several biomolecules, including vitamins, yeasts, enzymes, algae, biodegradable polymers, and microbes as well as waste plant components, have been successfully used in green synthesis techniques to create nanoparticles [3,10].

Singlet oxygen, superoxide anion, hydroxyl radical, and other extremely harmful reactive oxygen species (ROS) are produced in cells during normal metabolic activity. Some life functions, such as signal transduction and the creation of energy to power biological processes, need the presence of ROS; nevertheless, at excessive concentrations, ROS can damage macromolecules such as DNA, proteins, and lipids [11].

Researchers discovered a relationship between oxidative damage and the pathogenesis of oxidative illnesses such as cancer, vascular issues, and diabetes. Antioxidants have been shown to have a significant function in protecting against various illnesses. Antioxidants can delay or postpone the start or propagation phases of an oxidative chain reaction, thus preventing ROS-mediated damage. AgNPs inactivate the respiratory chain in mitochondria by generating reactive oxygen species (ROS). In this way, they can also damage cancer cell DNA and inhibit cell proliferation by inducing apoptosis [12]. Cancer is estimated to be the cause of one out of every six deaths worldwide. According to reports, 10 million individuals died as a result of cancer-related causes in 2020 [13]. As is well known, almost all cancers (90–95%) are caused by gene changes under the impact of environmental factors. Most recently, interest in AgNPs has been increasing, especially due to their antimicrobial [14], antioxidant [15], and anticancer [8] activities.

In addition, the increase in the resistance of bacteria to antibiotics due to their excessive/incorrect consumption is considered a sign of the beginning of the end for existing antibiotics that are widely used around the world. AgNPs, which have the potential to be used instead of antibiotics, can interact with the cell membrane surface and penetrate the bacterial cell membrane [16].

Alzheimer’s disease (AD), a fatal neurodegenerative condition that is progressive in nature, has emerged as a significant public health issue, particularly in industrialized nations where living standards are higher. It is also a common form of dementia, particularly among the elderly population, and is characterized by irreversible neuronal loss and abnormal behavioral changes. Psychosocial interventions, disease-modifying therapies, psychiatric drugs, and particularly cholinesterase inhibitors that prevent the hydrolysis of the two chemical neurotransmitters acetylcholine (AChE) and butyrylcholine are among the treatments for AD (BuChE). Numerous research has been carried out demonstrating the potential of plant-based synthetic metallic nanoparticles as anticholinesterase drugs [17]. Many studies have been conducted showing that plant-based synthesized metallic nanoparticles can be used as anticholinesterase agents [17,18,19,20,21].

*A. cepa*, also known as onion, Egyptian onion, and shallot, is regarded as both a spice and a vegetable. Typically, onions come in a range of hues, including white, red, yellow, green, and purple [22]. The onion is cultivated as an annual or biennial and can reach a height of 15–45 cm. It has six-part white blossoms, glossy black seeds, and coiled yellowish-green leaves that are flat and fan-shaped [23]. Onions are rich in bioactive phyto compounds such as anthocyanins, flavonoids, phenolic acids, and organosulfur compounds [24]. *Allium* species were used in traditional/complementary medicine treatment of some diseases such as asthma, cough, and vascular disease due to the antioxidant, antiobesity, anti-inflammatory, antimicrobial, antidiabetic, cytotoxic, reproductive-protective, immunomodulatory, hepatorenal-protective, digestive system-protective, respiratory-protective, cardiovascular-protective, and neuroprotective effects of identified bioactive compounds [25]. There were many studies in which *Allium* species were used as coating and reducing agents due to their rich chemical compositions. In these studies, many biological activities were tested in metallic nanoparticles synthesis research with extracts obtained from different parts of *Allium* species using different solvents. These studies had shown that *Allium* species could be used as suitable coating and reducing agents for green synthesis studies [26,27,28,29,30,31]. Therefore, the current work intends to evaluate the potential of *A. cepa* aqueous yellow-colored peel extracts’ anticholinesterase, antibacterial, and antioxidant properties utilizing affordable, environmentally acceptable AgNPs.

## 2. Results and Discussion

### 2.1. Analysis of UV-Vis Spectroscopy

Using a UV-Vis spectrophotometer, it was possible to see the reaction that was brought on by the color shift in the mixture made from *Allium cepa* peel extract and AgNO_3_ solution. AC-AgNPs’ surface plasmon resonance (SPR), which occurs throughout the reaction period, caused a color shift from bright yellow to dark black (Figure 1). To find the highest wavelength in the 300–700 nm region, samples collected at intervals of 10, 15, and 30 min were scanned. A total of 438.90 nm was found to be the maximum wavelength of biogenic AgNPs. This value was found to be in accordance with the highest absorbance values for AgNPs generated from plant extracts of Calophyllum tomentosum (438 nm) [32], Gymnema sylvestre (442 nm) [33], Cocos nucifera (380 nm) [34], and Cicer ariet [35]. A 2.4 GHz laser beam was used to measure the colloidal solution’s light transmittance after the reaction.

### 2.2. FE-SEM, TEM, and AFM Analysis

To evaluate the size and shape of the produced metallic nanoparticles (MNPs), FE-SEM (Figure 2) and TEM (Figure 3) methods were utilized. The synthetic AgNPs appear to be mostly spherical. Numerous studies reported that AgNPs made from diverse biological sources ranged in size from 14 to 61 nm and were generally spherical [36,37,38].

The TEM was used to analyze the shape and size distributions of biosynthesized AgNPs. Separate silver particles and the polydispersity of nanoparticles were seen in the TEM images (Figure 3a–d). According to TEM metrics, the dimensions of the synthesized nanomaterial were measured as a minimum of 8.44 nm and a maximum of 19.93 nm (Figure 3). Depleting the *A. cepa* extract throughout the synthesis time resulted in a reduction in reducing agents, which resulted in varied sizes of AC-AgNPs. We observed that there was no direct contact between nanoparticles even inside the aggregates, indicating that AC-AgNPs were covered by a thin layer of functional material originating from organic compounds that were typical of the creation of metallic nanoparticles from plant extracts [39].

To analyze metallic nanoparticles in contact mode, AFM was used as a structural characterization tool. AFM’s surface pictures and spatial resolution at the lateral and vertical nanometer scales enable it to quantitatively give significant information about the specifics of the surface topography. AC-AgNPs’ two- and three-dimensional (2D-3D) topographic characteristics were identified with the use of AFM (Figure 4). Color changes in the scale bar provide information about the size of the particles.

### 2.3. EDX Analysis of the AC-AgNPs

The EDX spectrum (Figure 5) revealed strong signals indicating the presence of Ag atoms in the biosynthesized nanomaterial. The presence of carbon, chlorine, and oxygen elements in the spectrum, which were sources of weak signals, was due to the plant extract. AgNPs showed a characteristic optical absorption peak at about 3 KeV dependent on SPR, as reported in many studies with different plants [3,40,41].

### 2.4. X-ray Diffraction (XRD) Analysis of A. cepa-AgNPs

Figure 6 shows the XRD spectrum pattern of AC-AgNPs. According to the spectrum data, the diffraction peaks at 37.89°, 44.02°, 64.25°, and 77.04°, which represent the cubic crystal structure of silver (Ag JCPDS No. 65–2871) at 2θ, index the planes (111), (200), (220), and (311), respectively. The peaks representing the crystal structure of silver were reported in many herbal silver nanoparticle synthesis studies such as *Cinnamomum camphora* [42], *Crossopteryx febrifuga, Brillantaisia patula, Senna siamea* [43], *Cicer arietinum* [44], *Prunus dulcis* [45], *and Allium ampeloprasum* [46]. The peak angle of 37.89 (highest peak) was taken to calculate the average particle size of AC-AgNPs. It was calculated to be around 19.47 ± 1.12 nm using the Debye–Scherrer equation (D = Kλ/(β cosθ)).

### 2.5. FT-IR Analysis

FT-IR spectroscopy was used to identify the functional groups responsible for the synthesis and stability of AgNPs produced during biosynthesis (Figure 7). It is clear that a variety of functional groups were engaged in the reduction of Ag^+^ ions when the frequency range of 4000–650 cm^−1^ shifts in the spectra of *A. cepa* peel aqueous extract (Figure 7a) and AC-AgNPs (Figure 7b) were examined. The bands at 1635 cm^−1^ in the FT-IR spectrum may represent ketone groups due to -C=O stretching, alkynes due to C-C stretching, and alcohol or phenol groups due to -OH stretching in the bands at 2134 cm^−1^ and 3317 cm^−1^, respectively [3,47].

### 2.6. TG and DT Analyze

AgNPs’ heat resistance as well as the weight loss that occurs when being heated to various temperatures were both determined using TG analysis (Figure 8). This method was used to determine the total amount of phytochemical residues present on AgNPs’ surface [48,49]. Furthermore, DT analysis indicated temperature fluctuations caused by exothermic/endothermic processes (Figure 8b). According to the findings, a mass loss of 2.5% happened in the sample at temperatures ranging from 24 °C to 250 °C, a mass loss of 5.5% occurred at temperatures ranging from 250 °C to 425 °C, and an 8.5% mass loss occurred at temperatures ranging from 425 °C to 889 °C (Figure 8a). These mass losses imply that the nanomaterial degrades slowly, indicating that the AgNPs generated are stable and durable even at high temperatures.

### 2.7. Zeta Size and Potential of AC-AgNPs

AC-AgNPs exhibited an average particle size of 134.6 nm and a zeta potential of −13.1 mV (Figure 9 and Figure 10). The negative zeta potential of AgNPs shows that the material was stable and that the particles did not adhere to one another [50,51]. Researchers reported studies that support our findings: *Origanum vulgare* (136 nm, −26 mV) [52], *Andrographis paniculata* (68.06 nm, −21.4 mV) [53], *Convolvulus arvensis* (90.9 nm, −18.5 mV) [54] and *Matricaria chamomilla* (45.12 nm, −34 mV) [55].

### 2.8. Antimicrobial Activity

Silver ions produced at the micro level in the aqueous media are primarily responsible for AgNPs’ antibacterial capabilities [56]. One of the possible scenarios where plant-mediated AgNPs with ultrasmall sizes and large surface areas suppress the activities and proliferation of microorganisms is that AgNPs bind to negatively charged cell surfaces and change the physical and chemical properties of the cell membrane and wall, thereby weakening the basic functions of the cell such as permeability, electron transport, osmoregulation, and respiration. The second is that AgNPs prevent the formation of the organism by interacting with DNA, proteins, and cell components containing sulfur/phosphorus [55,56].

Salomoni et al. [56] confirmed that AgNPs bind to bacterial DNA and subsequently attach to bacterial ribosomes, where they inhibit DNA replication and induce bacterial death. The most constant theme of an antibacterial investigation is the relationship between positively charged ions on Ag nanoparticles and negatively charged bacteria, which may also kill the bacterium [9]. AC-AgNPs suppressed yeast and bacterial growth more effectively than AgNO_3_ (except *E. coli*) and antibiotics (Table 1). When compared to the standard antibiotics, biogenic AC-AgNPs had the strongest inhibitory effect against *B. subtilis* (0.0625 μg/mL) and *S. aureus* (0.125 μg/mL).

The effectiveness of AC-AgNPs on the growth of microorganisms used in the study was found to be more effective than the standard antibiotics tested. These values were calculated to be 8, 8, 2, 16, and 4 times more effective for *S. aureus*, *B. subtilis*, *E. coli*, *P. aeruginosa*, and *C. albicans*, respectively. AgNPs’ MIC values (μg/mL) against the growth of *E. coli* and *P. aeruginosa* bacterial strains and *C. albicans* yeast were 0.25, 1.00, and 0.50, respectively (Table 1). AgNPs were shown to be particularly efficient at inhibiting the growth of *B.subtilis* (0.0312 μg/mL) and *S. aureus* (0.125 μg/mL). In another investigation, AgNPs were shown to be more effective at inhibiting *S.aureus* (6.25–50 μg/mL) and *E.coli* (2.5 μg/mL) [57].

### 2.9. Antioxidant Activity of AC-AgNPs

The existence of distinct types of functional groups, as indicated in FTIR, may be responsible for the maximum activity of AgNPs. This shows that AgNPs may be employed as an alternative antioxidant in the treatment of disorders induced by free radicals. Many studies have shown that AgNPs derived from plant extracts have high antioxidant activity [58,59]. *A. cepa* includes a variety of chemicals that may donate hydrogen from their hydroxyl group (−OH) to free radicals, forming stable, highly reactive hydroxyl radicals [25].

Table 2 shows the antioxidant activity findings of AC-AgNPs assessed using five different techniques. As a result, when compared to the reference substances BHA, α-tocopherol, and EDTA, the nanoparticle demonstrated mild antioxidant activity. The antioxidant properties of AC-AgNPs were most powerful in the β-Carotene–linoleic acid test, the metal-chelating test, and the ABTS test, with IC_50_ values of 116.9 ± 1.20 µg/mL, 120.4 ± 0.98 µg/mL, and 128.5 ± 1.17 µg/mL, respectively.

### 2.10. Anticholinesterase Activity of Biosynthesized AgNPs

Alzheimer’s disease (AD), the most common kind of dementia, is characterized by oxidative stress and progressive neuronal degeneration and is typically accompanied by visible beta-amyloid deposits in the brain and low acetylcholine levels [60]. The most often used method in the management and treatment of AD is acetylcholinesterase (AChE) and butyrylcholinesterase (BChE) inhibitors such as donepezil, rivastigmine, and galantamine [61]. Table 3 displays the anticholinesterase activity of *A. cepa*-AgNPs. According to the data, the nanoparticle inhibited both enzymes to a small extent. *A. cepa*-AgNPs inhibited AChE and BChE with IC_50_ values of 87.250.56 µg/mL and 71.330.98 µg/mL, respectively.

## 3. Materials and Methods

### 3.1. Materials and Reagents

*A. cepa* peels used in the reduction reactions of AgNO_3_ salt were obtained from local markets in Diyarbakır (Turkey). The Microbiology Research Laboratory at Mardin Artuklu University provided the pathogenic bacterial strains (*Bacillus subtilis* ATCC 11774, *Staphylococcus aureus* ATCC 29213, *Escherichia coli* ATCC 25922, and *Pseudomonas aeruginosa* ATCC 27853) and yeast (*Candida albicans* ATCC 10231), which were utilized to test the growth-suppressing impact of produced AgNPs. Sigma Aldrich-Merck (KGaA, Darmstadt, Germany) was used to get the chemicals, reagents, and standard antibiotics (colistin for Gram-positive (*B. subtilis* and *S. aureus*) strains, fluconazole for yeast (*C. albicans*), and vancomycin for Gram-negative (*E. coli* and *P. aeruginosa*) strains) required for the testing of the antimicrobial, antioxidant, and anticholinesterase activities.

### 3.2. Extraction Process

Fresh *A. cepa* peels were thoroughly cleaned in deionized water and dried at 30 °C. An amount of 1000 g of dry leaves was mixed with 2 L of distilled water and 2 L of deionized water and boiled at 85 °C for 10 min. The *A. cepa* peel extract obtained was brought to room temperature. Then, it was filtered and kept in a cold environment (+4 °C) to be used in the biosynthesis of AgNPs.

### 3.3. Biosynthesis of AC-AgNPs

A 40-millimolar (mM) AgNO_3_ aqueous solution was prepared using solid AgNO_3_ to perform the green synthesis of AgNPs. At 40 °C, 200 mL of *A. cepa* peel extract and 200 mL of AgNO_3_ solution were mixed and allowed to react. The solution was centrifuged at the required speed and time (4500 rpm, 10 min). The solid phase accumulating at the bottom was washed many times with deionized water. It was dried for 12 h at 55 °C and pulverized in a mortar.

### 3.4. Structural and Thermal Characterization of AC-AgNPs

Identification of synthesized AgNPs was done with an Agilent CARY 60 model spectrophotometer (300–800 nm). Powder crystal structures, morphologies, surface distributions, sizes, and values of NPs were measured with XRD (Rigaku), EDX (Quanta FEG240), AFM (PARK NX10), FE-SEM (Quanta FEG240), TEM (HITACHI 7700), and Zetasizer (Malvern). The Debye–Scherrer equation was used to compute the powder crystal size of AgNPs [4].
D = Kλ/(β cosθ)
where D = consistent diffraction area size (nm), K = Scherer constant (K = 0.94), λ = wavelength, β = the reflection width (2θ).

FT-IR (Agilent Cary 630) analysis was applied to identify the groups causing the reduction in the extract. TG/DT analysis (Instruments Q600) was applied to determine the stability and durability of the synthesized nanoparticle.

### 3.5. Antimicrobial Activity or Antibacterial and Antfungal Activity of AC-AgNPs

To evaluate the antibacterial efficacy of AC-AgNPs, the modified MIC assay was applied [62]. The Mueller Hinton Broth medium for bacterial strains and RPMI 1640 for yeast in 96-well microplates were used. Afterward, the medium was mixed with the appropriate amounts of bacteria and AgNP solution. It took a day for the media to incubate. Measuring the concentration in the well where the growth started allowed for the MIC value to be calculated. AC-AgNPs’ antibacterial activity was assessed using the conventional antibiotics fluconazole, colistin, and vancomycin (128 μg/mL each) as well as a 5 mM AgNO_3_ solution.

### 3.6. Anticholinesterase Activities of AC-AgNPs

The Ellman technique, with slight changes, was used to measure the inhibitory potential of AC-AgNPs on cholinesterase enzymes (AChE and BChE) spectrophotometrically [63,64]. Amounts of 130 μL of sodium phosphate buffer (pH 8.0, 100 mM) and 10 μL of sample solution were combined in various quantities. The combination was mixed and incubated at 25 °C for 15 min before being added 20 μL of 0.5 mM DTNB (5,5′-dithiobis(2-nitrobenzoic) acid). This was done after adding 20 μL of enzyme (AChE or BChE) buffer solution. The reaction was started by adding 20 μL of the substrates acetylthiocholine iodide (0.71 mM) or butyrylthiocholine chloride (0.2 mM). Using a 96-well microplate reader, the absorbance and appearance of the yellow 5-thio-2-nitrobenzoate anion produced by the reaction of DTNB with thiocholine were determined at 412 nm. The outcomes were expressed as an enzyme inhibition percentage (percent) at a concentration of 200 g/mL and an inhibitory concentration for nanoparticles (IC_50_) of 50%.

### 3.7. Antioxidant Activities of AC-AgNps

The DPPH (2,2-Diphenyl-1-picrylhydrazyl) free radical scavenging activity was assessed spectrophotometrically [65]. The technique described by Erel [66], was used to assess the radical scavenging activity of ABTS+ cations. Using the method created by Apak et al. [67], the cupric reducing antioxidant capacity (CUPRAC) was calculated. Spectrophotometric analysis was used to determine the extracts’ metal-chelating activity for Fe^2+^ [68]. EDTA was used as the control chemical in a comparison of the metal-chelating activity. To analyze the ABTS+, DPPH, β-carotene-linoleic acid, and CUPRAC assays, the common antioxidants BHA and α-tocopherol were used. The data for antioxidant activity was calculated as the 50% inhibitory concentration (IC_50_).

## 4. Conclusions

This research revealed that AgNPs can be synthesized from *Allium cepa* (AC) peels, which are considered biological waste, in an ecologically acceptable, low-cost and simple method. Characterization studies were performed with UV-Vis, EDX, FE-SEM, TEM, AFM, FTIR XRD, and Zetasizer. The FE-SEM and TEM images demonstrated that the produced AgNPs were predominantly in spherical morphology. The particle sizes of *A. cepa*-AgNPs were determined to be 8.44–19.93 nm based on TEM images and 19.47 nm based on XRD data. AgNPs with a zeta potential of −13.1 mV were thought to be resolute and stable. It was found that *A. cepa*-AgNPs had a greater biocidal impact on *P. aeruginosa*, *B. subtilis*, and *S. aureus*. The nanoparticle from *A. cepa* was shown to have the potential to be employed as an antioxidant agent. In addition, the capacity of nanoparticles to inhibit AChE and BChE shows that AC-AgNPs could be used to manage Alzheimer’s disease. This was also the first investigation of the anticholinesterase activity of AC-AgNPs. Even though AgNPs have numerous features that make them an ideal candidate for novel and potential medicinal applications, their toxicity has subsequently become an area of attention. AgNPs are commonly advertised as very efficient antibacterial agents that are safe for healthy mammalian cells, but the toxicity of AgNPs is connected with their conversion to biological and environmental systems.

The findings showed that biologically synthesized AgNPs have the potential to be used in many fields such as the biomedicine, pharmaceutical, and food industries, and their in-depth investigation is of critical importance. Nevertheless, the potentially toxic effects of AgNPs should be recognized before use, and it is suggested that the appropriate concentration of silver be known so it can perform skillfully without damaging individuals and their surroundings.

Potential applications for silver nanoparticles in the medical industry include cancer therapies and fluorescence imaging, where the nanoparticles allow for targeted drug administration, higher bioavailability and prolonged drug release in target tissues, and improved drug stability. However, more research is required to explore the mechanism of plant-mediated AgNPs implicated in biological applications, which is currently unknown. Such research might provide precise knowledge of the workings and efficiency of AgNPs.

AgNPs can be used for the production of many commercial and biomedical products on a large scale, because this protocol is simple and economical and the raw material is cheap.

## Figures and Tables

**Figure 1 molecules-28-02310-f001:**
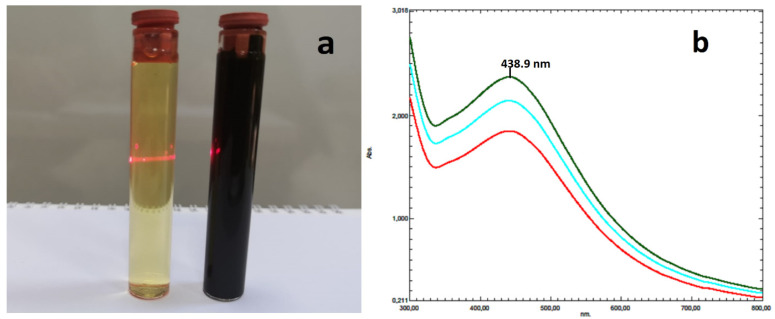
(**a**) Tyndall effect and visible laser beam path on colloidal AgNPs. (**b**) Time-dependent UV-Vis spectrum data of synthesized AgNPs.

**Figure 2 molecules-28-02310-f002:**
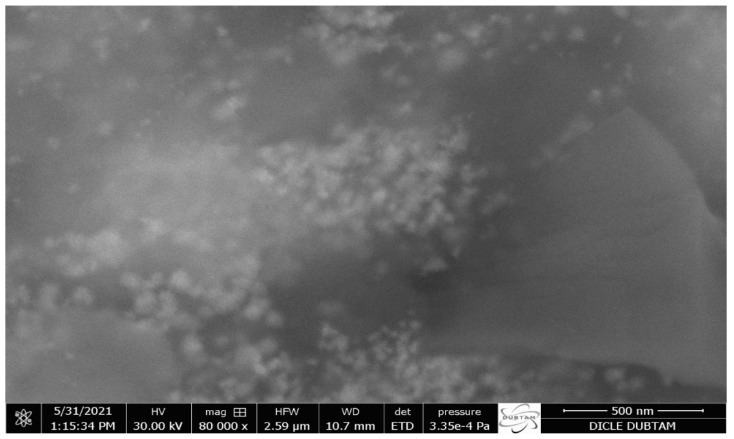
AC-AgNPs FE-SEM image at 500 nm scale.

**Figure 3 molecules-28-02310-f003:**
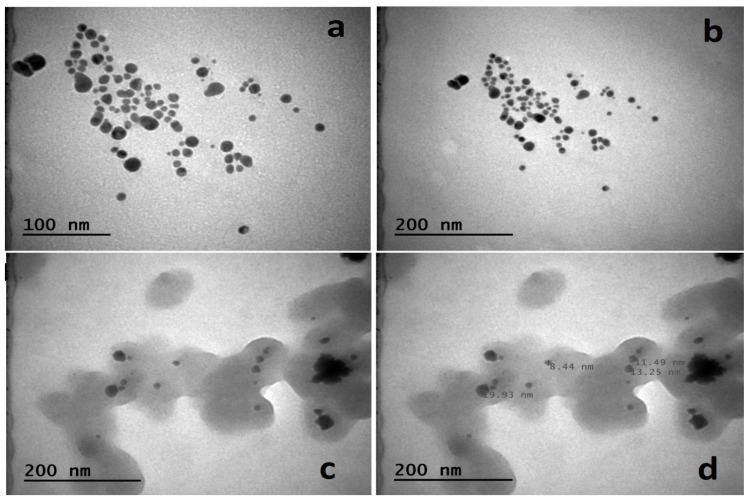
Unsized (**a**–**c**) and sized (**d**) TEM images of *A. cepa*-AgNPs.

**Figure 4 molecules-28-02310-f004:**
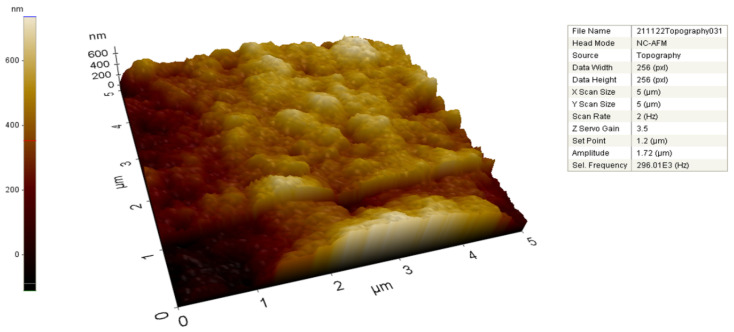
AFM image of AgNPs synthesized from *A. cepa* peel extract.

**Figure 5 molecules-28-02310-f005:**
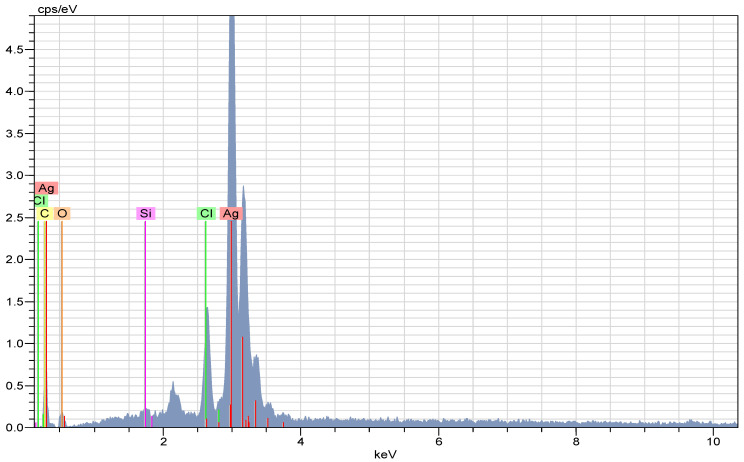
EDX spectrum demonstrating the presence of silver in AC-AgNPs.

**Figure 6 molecules-28-02310-f006:**
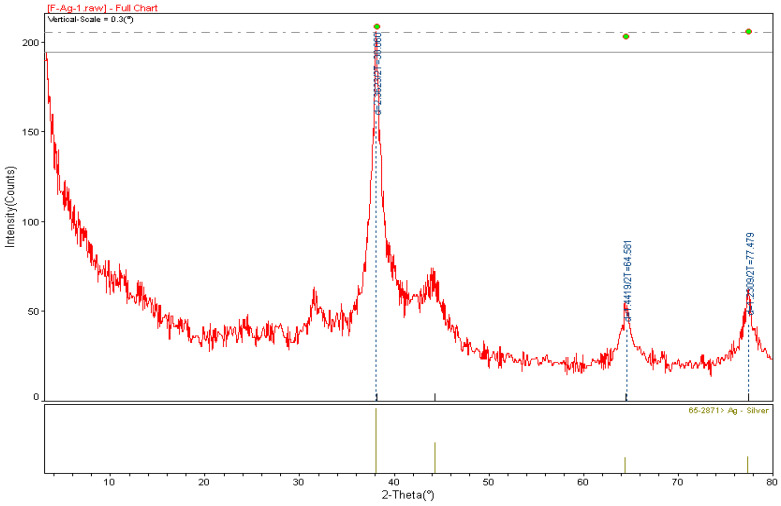
XRD pattern of AC-AgNPs powder crystal structure.

**Figure 7 molecules-28-02310-f007:**
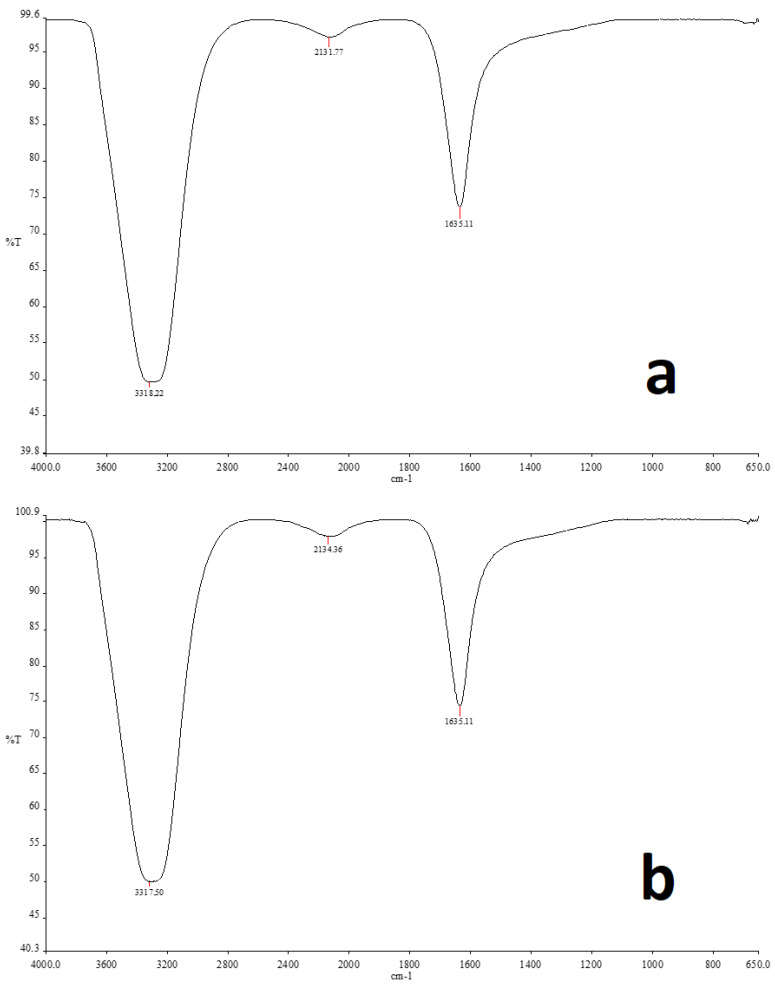
FT-IR spectra of *A. cepa* peel aqueous extract (**a**) and biosynthesized AgNPs (**b**).

**Figure 8 molecules-28-02310-f008:**
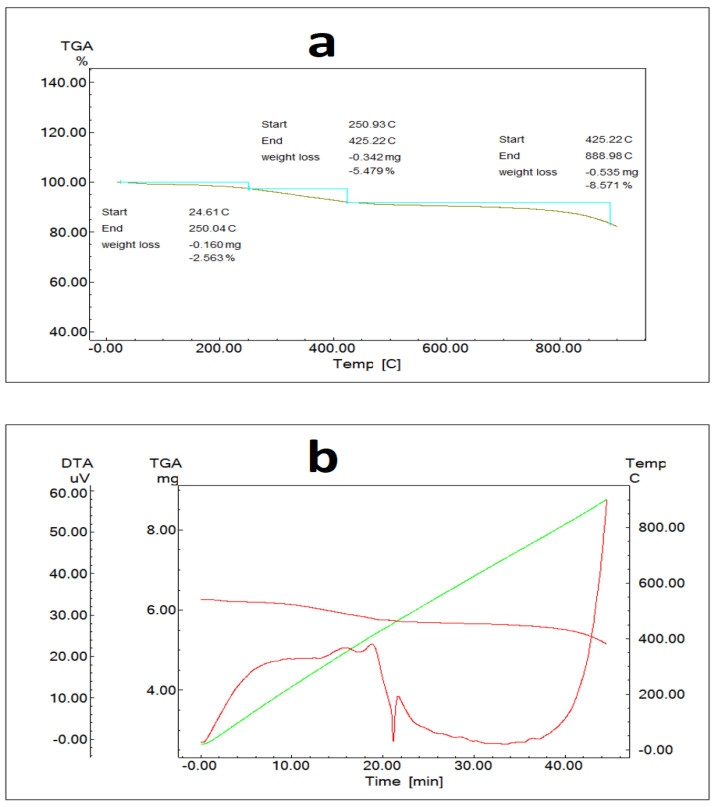
The findings of TG and DT analyses reveal that biosynthesized AgNPs were stable and durable at high temperatures (**a**,**b**).

**Figure 9 molecules-28-02310-f009:**
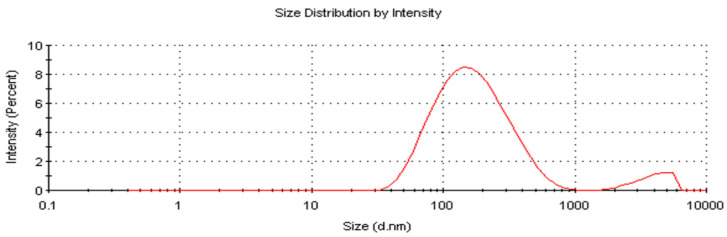
Size distribution pattern of AC-AgNPs.

**Figure 10 molecules-28-02310-f010:**
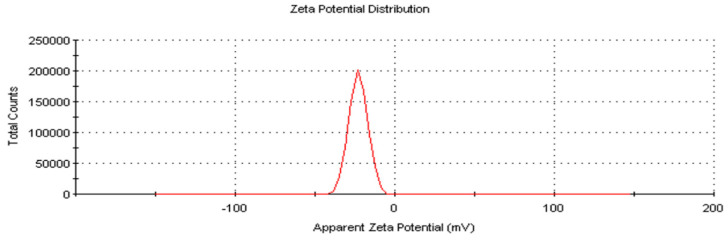
Zeta potential pattern of AC-AgNPs.

**Table 1 molecules-28-02310-t001:** MIC values (μg/mL) of AC-AgNPs, AgNO_3_ solution, and standard antibiotics.

Pathogenes	AC-AgNPs µg mL^−1^	AgNO_3_ Solution µg mL^−1^	Control Antibiotics µg mL^−1^ *
*Staphylococcus aureus* ATCC 29213	0.125 ± 0.1	2.65 ± 0.2	2.00 ± 0.2
*Bacillus subtilis* ATCC 11774	0.0625 ± 0.2	1.32 ± 0.1	1.00 ± 0.1
*Escherichia coli* ATCC 25922	1.00 ± 0.1	0.66 ± 0.2	2.00 ± 0.2
*Pseudomonas aeruginosa* ATCC 27853	0.25 ± 0.2	1.32 ± 0.2	4.00 ± 0.2
*Candida albicans* ATCC 10231	0.50 ± 0.1	0.66 ± 0.1	2.00 ± 0.1

* Colistin was used to treat Gram-positive (*B. subtilis* and *S. aureus*) strains, fluconazole was used to treat yeast (*C. albicans*), and vancomycin was used to treat Gram-negative (*E. coli* and *P. aeruginosa*) strains. All the treatments were performed in triplicate. Means and standard deviations were reported as mean ± SD.

**Table 2 molecules-28-02310-t002:** Antioxidant activity of AC-AgNPs as determined by DPPH•, ABTS•, CUPRAC, and metal-chelating assays (µg/mL).

Sample/Standard		β-Carotene/Linoleic Acid Assay	DPPH^•^ Assay	ABTS^•+^ Assay	CUPRAC Assay	Metal-Chelating Assay ^a^
		IC_50_	IC_50_	IC_50_	A_0_._50_	IC_50_
**Sample**	AC-AgNPs	116.9 ± 1.20	151.3 ± 1.65	128.5 ± 1.17	143.8 ± 1.67	120.4 ± 0.98
**Standards**	α-Tocopherol	2.10 ± 0.05	38.15 ± 0.45	35.50 ± 0.56	61.40 ± 0.75	NT ^b^
	BHA	1.45 ± 0.03	19.82 ± 0.33	12.80 ± 0.08	25.50 ± 0.43	NT
	EDTA	NT	NT	NT	NT	5.50 ± 0.45

^a^ The values represent the means ± SEM of three parallel sample measurements (*p* < 0.05). ^b^ NT: Not tested.

**Table 3 molecules-28-02310-t003:** The anticholinesterase activity of *A. cepa*-AgNPs.

	AChE		BChE	
Extracts/Standards	Inhibition (%) (at 200 µg/mL)	IC_50_ (µg/mL)	Inhibition (%) (at 200 µg/mL)	IC_50_ (µg/mL)
***A. cepa*-AgNPs**	62.20 ± 0.47	87.25 ± 0.56	66.27 ± 0.44	71.33 ± 0.98
**Galantamine**	85.50 ± 0.60	5.50 ± 0.20	74.65 ± 0.25	42.20 ± 0.45

The values represent the average ± SEM of three parallel sample measurements (*p* < 0.05).

## Data Availability

All data used to support the findings of this study are included in the article.

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
