# Peer review of "Green Synthesis of Silver Nanoparticles from *Allium cepa* L. Peel Extract, Their Antioxidant, Antipathogenic, and Anticholinesterase Activity"

_molecules, 2023, doi:10.3390/molecules28052310_

Round 1

Reviewer 1 Report

The work present on silver nanoparticles is worthy and green synthesis. The silver nanoparticles were synthesized by Allium cepa L. peel extract for biological application. However, the present structure, interpretation, and references of the work are not meet to the standard of the journal. I, therefore, feel that this manuscript can be published after the Major Revision.

Comments:

1.        Already, various works is published on silver nanoparticles, although why authors have chosen silver metal for the said study.

2.        Keywords should be replaced by eye-catching words.

3.        In the introduction authors stated “Various plant compo-51 nents including leaves, roots, and fruits, are generally used for biosynthesis applications, 52 also called “green synthesis”. The authors are encouraged to refer to the following recent works based on plant mediated nanoparticles synthesis-ACS Omega, 7 (2022) 6869−6884; Emerging Materials Research, 10 (2021) 75-84.

4.        Authors are appealed to incorporate some recent work which is on Allium cepa L. peel extract or the same family species that can act as a capping/reducing agent to fabricate Ag NPs.

5.         The plant, pathogens, and standard antibiotics names should be in italics. The abbreviation of nanoparticles is NPs, not NP. Make changes in the whole manuscript.  Moreover, there should be space between Ag and NPs.

6.        TEM should be discussed in detail by paying attention to particle nature. The particles show monodisperse behavior.

7.        Provide JCPDS no of the material in Fig. 6. Estimate crystallite size of material using Debye-Scherer formula.

In FTIR discussion, “The bands at 1635 cm-1 in the FT-IR spectrum may 227 represent ketone groups due to -C=O stretching, alkynes due to CC stretching, and alcohol 228 or phenol groups due to -OH stretching in the bands at 2134 cm-1 and 3317 cm-1, respec-229 tively (Wanjari et al., 2022; Atalar et al., 2021)”. How it is possible a stretching frequency obtained at 3317 cm-1 due to -OH/Water molecule. Because the sample was already dried at 100 ℃ for twelve hours. Just it modifies, it is due to the atmospheric water or contaminated during the analysis or air, water or CO2 background.

8.        Write TG instead of TGA. The discussion should be scientific.

9.        Please provide sufficient control experiments (+ve and –ve) to validate the biological data collected. Also, compare all pathogens' results with standard antibiotics. How many times run the sample.

10.     Highlight the novelty and findings of the work in the conclusion.

Suggestion: Major Revision

Author Response

Q1.        Already, various works is published on silver nanoparticles, although why authors have chosen silver metal for the said study.

R1. Nowadays, biosynthesis of silver nanoparticles (AgNPs) had gained so much attention in developed countries due to the development demand for environmentally friendly technology for material synthesis. The most important reasons why silver salts are preferred in nanoparticle synthesis for biological research are; The stability of silver nanoparticles is high, silver salts are relatively cheaper and accessible than other metal solutions (such as gold), and most importantly, they are compatible with biological systems. This novel green synthesis methodology offers a more cost-effective and environmentally responsible alternative to traditional methods for producing biocompatible nanoparticles. Green nanomaterials that have been created have the potential to be used in a variety of pharmaceutical applications, including the development of useful nanodevices and the development of novel pharmaceuticals.

Q2.        Keywords should be replaced by eye-catching words.

R2. Keywords were rearranged. Thanks.

Q3.        In the introduction authors stated “Various plant compo-51 nents including leaves, roots, and fruits, are generally used for biosynthesis applications, 52 also called “green synthesis”. The authors are encouraged to refer to the following recent works based on plant mediated nanoparticles synthesis-ACS Omega, 7 (2022) 6869−6884; Emerging Materials Research, 10 (2021) 75-84.

R3. Suggested works was added  to  the manuscript.

Q4.        Authors are appealed to incorporate some recent work which is on Allium cepa L. peel extract or the same family species that can act as a capping/reducing agent to fabricate Ag NPs.

R.4. We added new related studies to the introduction section.

Q5.         The plant, pathogens, and standard antibiotics names should be in italics. The abbreviation of nanoparticles is NPs, not NP. Make changes in the whole manuscript.  Moreover, there should be space between Ag and NPs.

R5. We corrected mentioned errors in the whole manuscript.

Q6.        TEM should be discussed in detail by paying attention to particle nature. The particles show monodisperse behavior.

R6. We added new text about the TEM in discussion part in page 7.“The TEM was used to analyze the shape and size distribution of biosynthesized AgNPs. Separate silver particles and the poly-dispersity of nanoparticles were seen in the TEM images (Fig. 3 a-d). According to TEM metrics, the dimensions of the synthesized nanomaterial were measured as a minimum of 8.44 nm and a maximum of 19.93 nm (Figure 3). Depleting the A. cepa extract throughout the synthesis time resulted in a reduction in reducing agents, resulting in varied sizes of AC-Ag NPs. We observed that there was no direct contact between nanoparticles even inside the aggregates, indicating that AC-Ag NPs were covered by a thin layer of functional originating from organic compounds which were typical of the creation of metallic nanoparticles from plant extracts (Jalilian et al., 2020)”.

Q7.        Provide JCPDS no of the material in Fig. 6. Estimate crystallite size of material using Debye-Scherer formula.

The bands at 1635 cm-1 in the FT-IR spectrum may 227 represent ketone groups due to -C=O stretching, alkynes due to CC stretching, and alcohol 228 or phenol groups due to -OH stretching in the bands at 2134 cm-1 and 3317 cm-1, respec-229 tively (Wanjari et al., 2022; Atalar et al., 2021)”. How it is possible a stretching frequency obtained at 3317 cm-1 due to -OH/Water molecule. Because the sample was already dried at 100 ℃ for twelve hours. Just it modifies, it is due to the atmospheric water or contaminated during the analysis or air, water or CObackground.

R7. Thank you for your valuable advices. We added requested information in page 9.  Ag JCPDS No. 65-2871 added in the text. The peak angle of 37.89 (highest peak) was taken to calculate the average particle size of AC-Ag NPs. It was calculated to be around 19.47 nm using the Debye-Scherrer equation. Post-synthesis drying was carried out at 55 °C for 12 hours. The incorrect expression was corrected in the text. The main reason green synthesis studies are preferred is phytochemicals, which are natural chemicals. These phytochemicals contain functional groups such as -OH, -C=O, and -N=N bonds. As long as these phytochemicals are in the reaction medium, they cause the formation of nanoparticles of various sizes. These formed nanoparticles are surrounded by these functional groups. Interaction with biological systems occurs through these functional groups. These functional groups are heat stable and do not degrade or become volatile even at high temperatures. The fact that the wavelengths measured at different time intervals with the UV-Spectrophotometer were the same after the reaction takes place, shows us that the synthesis was taken place and was stable. Also, TG and DT analyses showed us that the structures of the synthesized nanoparticles were largely preserved despite the temperature. Among the functional groups in the FTIR analysis, the -OH group may belong to the phenolic group in the group with 3340 cm-1. It can be 1635 cm -1 C=O or N=O group. FTIR analyses were performed in an aqueous medium and the average reaction temperature was 50°C. This temperature value was not a value that can make functional groups volatile. FTIR analysis was performed with the plant extract before the reaction and after mixing the plant extract and silver nitrate solution after the reaction. There was no exposure to high heat in any of these stages.

Q8.        Write TG instead of TGA. The discussion should be scientific.

R8. We edited according to your correction. We edited discussion based on the scientific results.

Q9.        Please provide sufficient control experiments (+ve and –ve) to validate the biological data collected. Also, compare all pathogens' results with standard antibiotics. How many times run the sample.

R9. We added requested information in Table 1. 

Q10.     Highlight the novelty and findings of the work in the conclusion.

R.10. Necessary improvements were made to the conclusion section.

Reviewer 2 Report

This manuscript investigated the Antioxidant, Antipathogenic, and Anticholinesterase effects of  Allium cepa L. peel extract based silver nanoparticles. This article claims that using of this type of silver nanoparticles could be a suitable for biomedical applications, as well as infection disease treatment. Therefore, I suggest a minor correction and require a detailed clarification. Correction to be addressed by the authors as follows: The abstract is not well organized, where the sentences are incomplete and no continuity is there. It would be feasible, if include the significance of the current study in the abstract. A brief description of how the authors selected information from the literature in the databases, as well as doses.
Authors should justify and expand the information on the biomedical application of green nanoparticles in which this species is mentioned, highlighting the main contribution in in vitro fields.
Authors should specify the main experimental conditions used on the evidences from the literature. Where they briefly describe the most important data reported in the literature in a homogeneous manner and sequence reinforcing the relevance of this species as medicinal alternative.
The most significant  mechanism of action of this nanoparticles should be described and noticed more emphatically. Authors should discuss whether the use of these nanoparticles represents a solid alternative to existing commercial drugs or a source of new drugs.
Please add more previous studies to your manuscript in discussion section and also please discuss about possible toxicity of proposed nanomaterials.
Conclusions should reaffirm the fundamental contribution of this paper.

Author Response

This manuscript investigated the Antioxidant, Antipathogenic, and Anticholinesterase effects of  Allium cepa L. peel extract based silver nanoparticles. This article claims that using of this type of silver nanoparticles could be a suitable for biomedical applications, as well as infection disease treatment. Therefore, I suggest a minor correction and require a detailed clarification. Correction to be addressed by the authors as follows:

Q1. The abstract is not well organized, where the sentences are incomplete and no continuity is there. It would be feasible, if include the significance of the current study in the abstract. A brief description of how the authors selected information from the literature in the databases, as well as doses.

R1. Suggestions were made in the abstract section.

Q2. Authors should justify and expand the information on the biomedical application of green nanoparticles in which this species is mentioned, highlighting the main contribution in in vitro fields.
Authors should specify the main experimental conditions used on the evidences from the literature.

Where they briefly describe the most important data reported in the literature in a homogeneous manner and sequence reinforcing the relevance of this species as medicinal alternative.

R2. Necessary evidence about the use of silver nanoparticles and the green synthesis process was added to the introduction. Please see page 3-4.

Q3. The most significant mechanism of action of this nanoparticles should be described and noticed more emphatically.

R3. Necessary additions were made to the results and discussion section.

Q4. Authors should discuss whether the use of these nanoparticles represents a solid alternative to existing commercial drugs or a source of new drugs. Please add more previous studies to your manuscript in discussion section and also please discuss about possible toxicity of proposed nanomaterials. Conclusions should reaffirm the fundamental contribution of this paper.

R4. Conclusion section was revised and necessary addition were made.

Round 2

Reviewer 1 Report

The revised version of the manuscript is fine. Now, I recommended to accept the article in its present form.     

Author Response

Thanks for constructive comments and positive comment of reviewer.

The final version of the article was checked by the English language expert of our University, and the suggested edits were highlighted in the text.